

# *In-silico* assay of a dosing vehicle based on chitosan-TiO₂ and modified benzofuran-isatin molecules against *Pseudomonas aeruginosa*

Verónica Castro-Velázquez[1,2], Erik Díaz-Cervantes[2,*],
Vicente Rodríguez-González[1,*] and Carlos J. Cortés-García[3]

[1] División de Materiales Avanzados, Instituto Potosino de Investigación Científica y Tecnología, San Luis Potosí, San Luis Potosí, Mexico
[2] Departamento de Alimentos, Universidad de Guanajuato, Tierra Blanca, Guanajuato, Mexico
[3] Laboratorio de Diseño Molecular/Instituto de Investigaciones Químico Biológicas, Universidad Michoacana de San Nicolás de Hidalgo, Morelia, Michoacán, Mexico
* These authors contributed equally to this work.

Corresponding authors
Erik Díaz-Cervantes, e.diaz@ugto.mx
Vicente Rodríguez-González, vicente.rdz@ipicyt.edu.mx

## ABSTRACT

A high priority of the World Health Organization (WHO) is the study of drugs against *Pseudomonas aeruginosa*, which has developed antibiotic resistance. In this order, recent research is analyzing biomaterials and metal oxide nanoparticles, such as chitosan (QT) and TiO₂ (NT), which can transport molecules with biological activity against bacteria, to propose them as drug carrier candidates. In the present work, 10 modified benzofuran-isatin molecules were studied through computational simulation using density functional theory (DFT) and molecular docking assays against Hfq and LpxC (proteins of *P. aeruginosa*). The results show that the ligand efficiency of commercial drugs C-CP and C-AZI against Hfq is low compared with the best-designed molecule MOL-A. However, we highlight that the influence of NT promotes a better interaction of some molecules, where MOL-E generates a better interaction by 0.219 kcal/mol when NT is introduced in Hfq, forming the system Hfq-NT (Target-NT). Similar behavior is observed in the LpxC target, in which MOL-J is better at 0.072 kcal/mol. Finally, two pharmacophoric models for Hfq and LpxC implicate hydrophobic and aromatic-hydrophobic fragments.

## INTRODUCTION

Among the main current problems are bacterial infections due to mutations, which create resistance to antimicrobial drugs because they can occur in different genome sites. In addition, they spread among the population by various mechanisms. The determining aspects of antimicrobial resistance can be classified into three groups: acquisition of foreign DNA, mutations of pre-existing genetic determinants, and mutations in acquired genes (*Mayers et al., 2009*); for example, mutations in lysin synthesis and amino acid transporters (*Moger-Reischer & Lennon, 2019*). In this order, the infection produced by the *P. aeruginosa* bacterium (target of research and development since 2017 by the WHO)

leads to epithelial damage, inhibition of wound repair, and dissemination of surrounding tissues and organs (*Ołdak & Trafny, 2005*; *Yayan, Ghebremedhin & Rasche, 2015*; *Tang et al., 2018*).

Due to antibiotic resistance, newer treatment approaches are more aggressive than previous treatments, knowing that the complex interactions between *P. aeruginosa* and the host impact the survival of patients infected with *P. aeruginosa*. Therefore, it is necessary to identify the host proteins and the active sites of the bacteria, which can be used as new pharmacological targets to generate resistance, having a substantial change concerning conventional antimicrobial therapies (*Molina-Santiago et al., 2016*). In this order, some molecular targets for *P. aeruginosa* are shown below.

First, the enzyme LpxC (UDP-3-O-(acyl)-N-acetyl glucosamine deacetylase) is involved in the pathophysiology and pathogenesis that induces human diseases (*Zhou & Barb, 2008*). With its co-factor (zinc), this enzyme produces lipid A biosynthesis, crucial in its catalytic and structural function. Above promotes the growth of gram-negative bacteria such as *P. aeruginosa*. Also, LpxC is vital for its regulatory role in lipid biosynthesis. In addition, it does not share a structural sequence with any mammalian protein, so that a particular inhibitor can be developed with little off-target affinity and toxicity (*Jackman et al., 2000*; *Coggins et al., 2003*; *Zhou & Hong, 2021*).

Another important target is the Hfq protein, which is known as a factor required for bacterial RNA replication. Its function is the regulation and modulation of the stability of the bacteria, facilitating the interaction between non-coding RNAs (RNAs) and messenger RNAs (mRNA). For the elimination of the Hfq gene, pleiotropic phenotypes have been used, increasing the sensitivity of the protein to UV rays, which causes oxidative stress and, consequently, decreases the rate of bacterial growth (*dos Santos, Arraiano & Andrade, 2019*).

Generally, the structures used to develop blocking, and inhibitory molecules of bacterial proteins have heterocyclic structures to improve their effectiveness (*Schâfer-Korting, 2010*; *Uvarov & Popov, 2013*), like fluoroquinolones (*Hossain, 2018*). The antimicrobial activity of indole derivatives or analogs has been developed by modifying their ring substituents and functionalizing them with other compounds to improve their effectiveness (*Manna & Agrawal, 2009*; *Renuka et al., 2014*; *Khodarahmi et al., 2015*). Benzofuran (*Sagaama et al., 2020*) and isatin are aromatic heterocycles of biological and pharmacologically relevance being considered privileged structures, due their derivates display biological properties including antimicrobial, antiparasitic, analgesic, anti-tubercular, antitumor, antifungal and cytotoxic activities (*Khanam & Shamsuzzaman, 2015*; *Nevagi, Dighe & Dighe, 2015*; *Guo, 2019*; *Chauhan et al., 2021*; *Farhat et al., 2022*). Among the biological activities mentioned above, these heterocycles have been used as potent microbial agents against *Pseudomonas aeruginosa*, see Fig. 1 (*Liu et al., 2012*; *Hiremathad et al., 2015*; *Al-Wabli, Zakaria & Attia, 2017*). We highlight that their conformation suggests their application as inhibitory antibacterial agents where their hydrophobic capacity can block the DNA gyrase of the bacteria (*Tisovský et al., 2017*; *Chemchem et al., 2020*).

On the other hand, $TiO_2$ is used as a dosage vehicle of biological-interest molecules (*Aguilera-Granja, Aguilera-del-Toro & Díaz-Cervantes, 2022*), as the delivery of drugs that

**R (substituents)**

**Mol-A**

1-methoxy-4-methylbenzene

**Mol-F**

1-fluoro-4-methylbenzene

**Mol-B**

1-fluoro-2-methylbenzene

**Mol-G**

1-chloro-3-methylbenzene

**Mol-C**

1-bromo-2-methylbenzene

**Mol-H**

1,2,3,4,5-pentafluoro-6-methylbenzene

**Mol-D**

1-chloro-2-methylbenzene

**Mol-I**

1-fluoro-3-methylbenzene

**Mol-E**

1,2-difluoro-4-methylbenzene

**Mol-J**

toluene

**Figure 1  Designed benzofuran (red structure)-isatin (blue structure) molecules.** R is the substituent by each molecule (A–J). The Hex code for the red color is #F8130B.

promote the inhibition of SARS-CoV-2 main protease, a perspective to controlled drug administration. Therefore, pH-sensitive polymers have been employed to overcome the problem of the release of $TiO_2$ drugs. In the case of the administration of temperature-sensitive drugs, nanoparticles have been used, exploring polymers and

temperature-sensitive drugs to improve their properties by decorating them with $TiO_2$ (*Lai et al., 2016*; *Rodríguez-González, Terashima & Fujishima, 2019*; *Rodríguez-González et al., 2020*).

In the case of chitosan, it is reported that as a positively charged molecule, chitosan interacts efficiently with cell membranes, which are negatively charged due to ion exchanges between the intracellular and extracellular medium, so cells usually absorb chitosan nanoparticles (*Ravi Kumar, 2000*; *Mukhopadhyay et al., 2018*). On the other hand, some theoretical assays of this molecule have been reported in the literature (*Kazachenko et al., 2021*). When formulating chitosan as a dosing vehicle, it is combined with a drug, where the number of positively charged amino groups that become available decreases comparatively (*Sudarshan, Hoover & Knorr, 1992*). Controlling the number of amino groups is paramount because their availability can affect their ability to interact with cell membranes and the surrounding environment, potentially decreasing their adsorption and triggering potential toxicity events (*Kim, 2018*).

## COMPUTATIONAL METHODS

### Geometrical optimization

The structures of the studied molecules see Fig. 1, were optimized using the density functional method (DFT) (*Arai et al., 2007*), though the software Gaussian (G09) (*Rappoport & Furche, 2010*), using the experimental-correlated generalized gradient approximation hybrid functional proposed by *Zhao & Truhlar, (2008)* (M06-2x), with a Pople basis set with polarizing and diffuse functions (6-31+g (d, p)) (*Yang, Parr & Pucci, 1984*; *Geerlings et al., 2014*). The frequencies of all optimized molecules were calculated to verify that they were a minimum at the potential energy surface (PES).

### Fukui condensed functions

Fukui condensed functions were used (*Fukui & Pullman, 1980*), Eqs. (1)–(3), to calculate the reactivity index of the molecules, as well as to know the reactive sites (areas where the molecules would interact more easily with another). These functions are based on the formalism of the conceptual DFT; in the present work, we obtained them from the analysis of natural orbitals (NBO) (*Fukui & Pullman, 1980*; *Yang, Parr & Pucci, 1984*; *El Ouafy et al., 2020*).

$$f_j^+ \equiv q_j(N+1) - q_j(N) \qquad (1)$$

$$f_j^- \equiv q_j(N) - q_j(N-1) \qquad (2)$$

$$f_j^0 \equiv \frac{q_j(N+1) - q_j(N-1)}{2} \qquad (3)$$

A

ciprofloxacin

B

azithromycin

C

4-chlorophenyl (5 -nitrobenzofuran 2-yl) carbamate

D

4-chlorophenyl (5-bromobenzofuran-2-yl) carbamate

E

(Z)-4-methoxy-3-((2-oxoindolin-3-ylidene) amino) benzoic acid

F

(Z)-3-((5-bromo-2-oxoindolin-3-ylidene) amino)-4-methoxybenzoic acid

**Figure 2  Control molecules.** Commercial drugs: (A) C-CIP, and (B) C-AZI; derivatives of benzofuran (*Budhwani, Sharma & Kalyane, 2017*): (C) BRA, and (D) BRB; derivates of isatin: (E) IRA, and (F) IRB.

## Molecular docking and pharmacophoric model

The binding interactions between ligands (studied molecules) and a macromolecular target (typically proteins), whose structure is experimentally known, were analyzed.

The technique involves locating the most likely binding between the ligand and the receptor with lower energy (the lower the energy, the stronger the binding) and the suitable molecular binding site (*Huggins et al., 2019*; *Lin, Li & Lin, 2020*).

The targets were selected considering their biochemical importance of them when interacting with ligands, promoting bacterial inhibition. The cartesian coordinates of the targets (Hfq and LpxC) were obtained from the protein data bank with PDB codes: 5I21 (*Roy et al., 2010*; *Lekontseva et al., 2020*) and 4J3D (*Hale et al., 2013*), respectively.

Water molecules and coupled ligands were deleted for both proteins, and only the cofactor responsible for their antibiotic action was taken as a reference. For the calibration of each protein the quadratic deviation values (RMSD) were evaluated and considered an RMSD less than 1 Å, specifically 0.5580, and 0.0017 for 4J3D, and 5I21 proteins respectively. Chimera software was used to add charges, remove solvents, and correct the structure to be analyzed (*Yang & Chen, 2004*).

*In silico* molecular coupling was performed between the targets described above and the 10 molecules studied (Fig. 1), as well as with the biomaterials studied: $TiO_2$ nanotube model (NT) and chitosan model (QT). To compare the ten studied molecules were added some control molecules (Fig. 2): two commercial drugs (C-Cip: ciprofloxacin, and C-Azi:

Table 1 Reactivity indices.

| Compound | F⁺ | | | F⁻ | | | F⁰ | | |
|---|---|---|---|---|---|---|---|---|---|
| MOL-A | C | 18 | 0.022 | C | 29 | 0.337 | C | 20 | 0.138 |
| | C | 26 | 0.015 | C | 20 | 0.331 | C | 29 | 0.128 |
| | N | 21 | 0.008 | H | 24 | 0.229 | H | 24 | 0.110 |
| | C | 13 | 0.007 | H | 31 | 0.222 | H | 31 | 0.101 |
| | C | 4 | 0.001 | H | 11 | 0.188 | H | 11 | 0.093 |
| MOL-B | C | 17 | 0.022 | C | 28 | 0.337 | C | 19 | 0.141 |
| | C | 25 | 0.015 | C | 19 | 0.336 | C | 28 | 0.128 |
| | N | 20 | 0.008 | H | 23 | 0.229 | H | 23 | 0.110 |
| | C | 12 | 0.007 | H | 30 | 0.222 | C | 5 | 0.101 |
| | C | 4 | 0.001 | C | 5 | 0.202 | H | 30 | 0.101 |
| MOL-C | C | 17 | 0.022 | C | 28 | 0.337 | C | 19 | 0.141 |
| | C | 25 | 0.014 | C | 19 | 0.336 | C | 28 | 0.128 |
| | N | 20 | 0.008 | H | 23 | 0.229 | H | 23 | 0.110 |
| | C | 12 | 0.007 | H | 30 | 0.222 | H | 30 | 0.101 |
| | C | 4 | 0.001 | H | 10 | 0.187 | H | 10 | 0.093 |
| MOL-D | C | 17 | 0.022 | C | 28 | 0.337 | C | 19 | 0.141 |
| | C | 25 | 0.015 | C | 19 | 0.336 | C | 28 | 0.128 |
| | N | 20 | 0.009 | H | 23 | 0.229 | H | 23 | 0.110 |
| | C | 12 | 0.007 | H | 30 | 0.222 | H | 30 | 0.101 |
| | C | 4 | 0.000 | H | 10 | 0.187 | H | 10 | 0.093 |
| MOL-E | C | 17 | 0.022 | C | 7 | 0.036 | C | 7 | 0.018 |
| | C | 25 | 0.015 | C | 4 | 0.017 | C | 25 | 0.009 |
| | N | 20 | 0.008 | C | 12 | 0.008 | C | 4 | 0.009 |
| | C | 12 | 0.007 | C | 25 | 0.002 | C | 12 | 0.007 |
| | C | 4 | 0.001 | N | 22 | 0.002 | N | 20 | 0.002 |

azithromycin, Figs. 2A, and 2B respectively); two benzofuran derivatives (*Budhwani, Sharma & Kalyane, 2017*), see Figs. 2C, and 2D; and two isatine derivatives (*Chemchem et al., 2020*), see Figs. 2E, and 2F. The molecular docking was carried out through the MolDock Score (*Thomsen & Christensen, 2006*) implemented in the molegro virtual docker package (MVD) (*Yang & Chen, 2004*).

This study compared which biomaterial could be introduced to the selected target to understand the best binding mode of a possible antibiotic-carrying nanovector.
We evaluated the above by first coupling the NT, then converting NT into a "cofactor" of the protein (in other words, considering it as part of the protein) so that the software would consider it part of the complete system and continue with the analysis. QT coupling on the said white-NT system. With this, we could corroborate how the presence of NT affects this type of nanodevice and the energy contribution due to the presence of said NT.

Converting the NT as a "cofactor" was also performed for the QT to couple the NT subsequently. Once this target-NT-QT or target-QT-NT system, the ten studied molecules

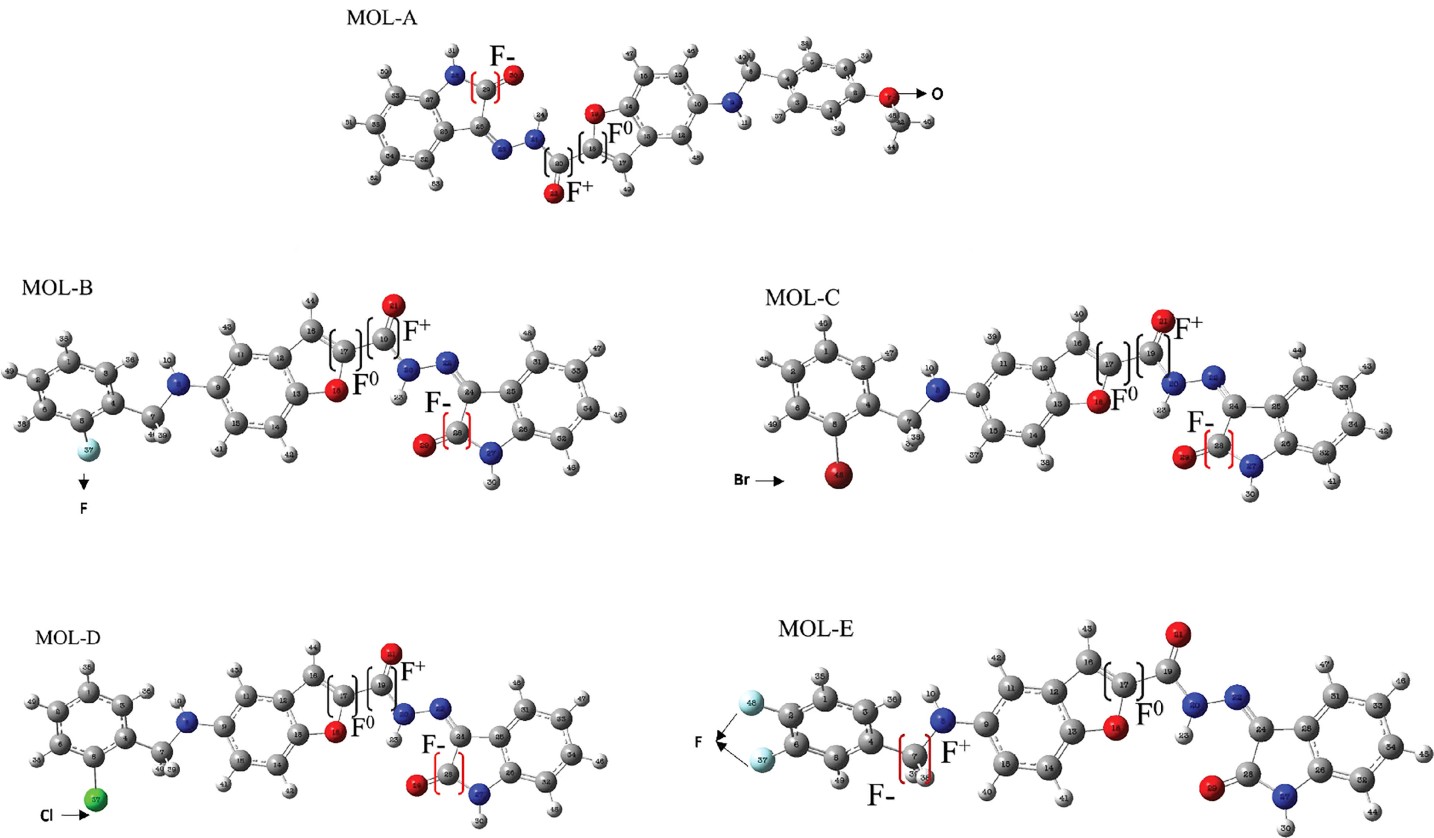

**Figure 3 Reactive sites of the first five designed molecules.** $F^+$ means nucleophilic attack more suitable site, $F^-$ is the more suitable site for an electrophilic attack, and $F^0$ is the more suitable radical-attack site. The Hex code for red color is #F40303 and the color green is #10EE11.



were coupled to evaluate the effect of the NT-QT or QT-NT type nanotransporter on a Pseudomonas target.

Finally, was proposed a pharmacophore model for each studied target considered the docking interactions. This study was made using the ZincPharmer (*Koes & Camacho, 2012*).

## RESULTS

### Condensed Fukui functions

All the studied molecules were optimized following the described method. The most susceptible zones of electrophilic, nucleophilic, and radical attacks were computed using the Fukui condensed functions. The carbonyl carbon linked to the benzofuran moiety is a high-reactivity atom for a nucleophilic attack for the designed molecules. The carbonyl carbon in the isatin fragment presents the most electrophilic-attack site. We highlight that these results are essential due to the fact that they can predict the most suitable sites of interaction of the benzofuran-isatin designed molecules with the NT and chitosan; see Table 1 and Fig. 3.

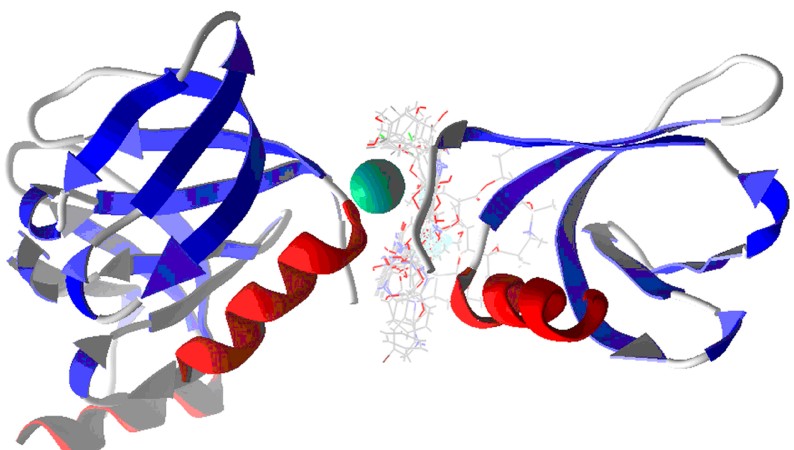

**Figure 4 Docking of studied molecules in Hfq protein.** The Hex code for red color is #FF2525 and the color green is #00744A.               

**Table 2 Ligand efficiency (kcal/mol) for Hfq protein: PDB code: 5I21 cavity: volume 25.600 A$^3$.**

| Name | Target | Target-NT | Target-QT | Target-NTQT | Target-QTNT |
|---|---|---|---|---|---|
| BRA | −5.443 | −5.234 | −4.783 | −3.868 | −4.083 |
| BRB | −5.138 | −5.166 | −4.340 | −3.892 | −4.355 |
| C-AZI | −2.126 | −2.088 | −2.055 | −1.703 | −1.882 |
| C-CIP | −3.744 | −4.318 | −3.830 | −2.699 | −3.670 |
| IRA | −4.676 | −5.051 | −3.760 | −3.116 | −4.377 |
| IRB | −5.317 | −5.079 | −4.403 | −3.081 | −4.369 |
| MOL-A | −4.265 | −4.067 | −3.180 | −3.117 | −3.039 |
| MOL-B | −4.123 | −3.902 | −3.328 | −3.204 | −3.079 |
| MOL-C | −4.229 | −3.919 | −3.475 | −3.406 | −3.182 |
| MOL-D | −4.211 | −3.930 | −3.473 | −3.301 | −2.949 |
| MOL-E | −4.102 | −4.321 | −3.192 | −3.165 | −2.993 |
| MOL-F | −4.127 | −4.054 | −3.327 | −3.540 | −3.081 |
| MOL-G | −4.107 | −4.045 | −3.537 | −3.571 | −3.100 |
| MOL-H | −3.690 | −3.310 | −2.678 | −2.742 | −2.679 |
| MOL-I | −4.153 | −4.078 | −3.525 | −3.538 | −3.248 |
| MOL-J | −4.102 | −4.084 | −3.446 | −3.475 | −3.284 |
| NT | −4.010 | – | −4.121 | – | – |
| QT | −3.454 | −3.307 | – | – | – |

## Molecular docking and pharmacophoric model

The docking was made in Hfq and LpxC targets, which are proteins in *P. aeruginosa*. Note that the main cavities of these targets were evaluated, see the Supplemental File, but this section only shows the better-docked cavity.

Figure 4 depicts the studied molecules docked with Hfq. Table 2 shows the ligand efficiency for the ten designed molecules, the referencing molecules, the model of QT, NT,

| Table 3 Main interacting energies for the studied molecules with Hfq-NT system. | | | | |
|---|---|---|---|---|
| Name | VdW | Electro | H-bond* | LE |
| C-AZI | −0.583 | 4.823 | 0.000 | −2.088 |
| QT | −1.041 | −0.666 | −7.032 | −3.307 |
| MOL-H | −1.936 | −1.714 | −7.976 | −3.310 |
| MOL-B | −2.333 | −2.156 | −5.590 | −3.902 |
| MOL-C | −2.270 | −2.118 | −5.668 | −3.919 |
| MOL-D | −2.092 | −2.266 | −7.381 | −3.930 |
| MOL-G | −2.250 | −2.100 | −5.966 | −4.045 |
| MOL-F | −2.129 | −1.991 | −7.209 | −4.054 |
| MOL-A | −2.058 | −2.129 | −7.354 | −4.067 |
| MOL-I | −2.260 | −2.123 | −3.981 | −4.078 |
| MOL-J | −0.173 | −0.256 | 0.000 | −4.084 |
| C-CIP | −1.957 | −1.214 | −3.900 | −4.318 |
| MOL-E | −0.182 | −0.003 | 0.000 | −4.321 |
| IRA | −0.634 | −1.643 | −5.000 | −5.051 |
| IRB | 1.421 | −1.135 | −9.374 | −5.079 |
| BRB | −1.416 | 1.172 | 0.000 | −5.166 |
| BRA | −1.924 | −0.219 | −2.500 | −5.234 |

**Note:**

* H-bond means the hydrogen bond interactions, electro is the electrostatic interactinos, VdW is the Van der Waals energy, and LE is the ligand efficiency (LE = E/No heavy atoms).

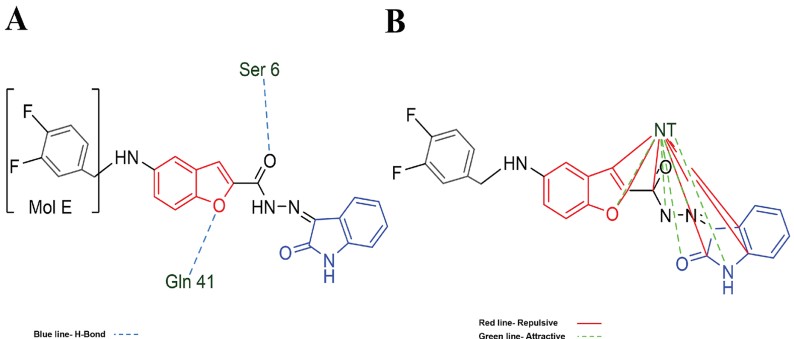

**Figure 5 MOL-E interacting with Hfq-NT (Target-NT) system.** (A) H-bonds, and (B) electrostatic interactions. Blue dotted lines depict the H-bond interactions, green dotted lines are the attractive, and red dotted lines are the repulsive electrostatic interactions. The Hex code red color is #F50708 and for green colors are #50FF4F and #034F01.

and the commercial drugs. Table 2 shows the ligand efficiency for the above molecules with the free target in the second column. The third column depicts the target with the NT linked, in other words, the system Target-NT is a simulation of the first introduction of NT in the target after testing the studied molecules. The four to six columns present similar behavior to the third, denoting the influence of QT, NTQT, and QTNT when this is docked in the target.

In this order, Table 2 shows that when introduced QT in the protein, see column four, the ligand efficiency of the systems is higher, which means a less interaction ligand-target.

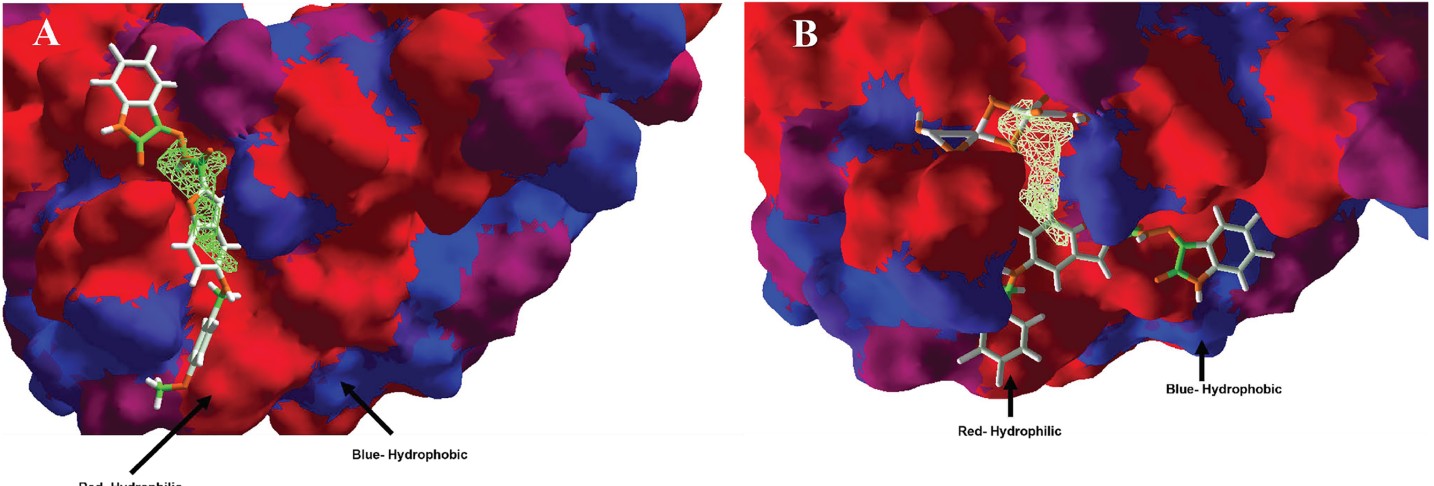

**Figure 6 Hydrophobic interactions between protein Hfq with (A) MOL-A, and Hfq-NT with (B) MOL-E.** Blue surfaces represent hydrophobic sites, and red surfaces are hydrophilic zones. The Hex code for red color is #FE0021, and for green colors are #4BE34B and #A1D788.

**Table 4 Efficiency of ligand (kcal/mol), for LpxC protein: PDB code: 4J3D cavity: volume 289.792 A$^3$.**

| Name | Target | Target-NT | Target-QT | Target-NTQT | Target-QTNT |
|------|--------|-----------|-----------|-------------|-------------|
| BRA | −4.709 | −4.919 | −5.169 | −5.107 | −5.284 |
| BRB | −4.834 | −4.825 | −4.847 | −5.263 | −4.664 |
| C-AZI | −2.717 | −2.487 | −2.332 | −2.526 | −2.519 |
| C-CIP | −4.273 | −4.272 | −3.838 | −4.251 | −4.155 |
| IRA | −5.116 | −4.868 | −4.833 | −4.745 | −4.915 |
| IRB | −5.101 | −4.665 | −4.787 | −4.776 | −4.745 |
| JS_303 | −4.794 | −4.693 | −4.460 | −4.502 | −3.921 |
| MOL-A | −4.378 | −4.635 | −4.215 | −4.645 | −4.433 |
| MOL-B | −4.638 | −4.815 | −4.322 | −4.368 | −4.067 |
| MOL-C | −4.867 | −4.903 | −4.567 | −4.563 | −4.370 |
| MOL-D | −4.883 | −4.819 | −4.403 | −4.263 | −4.320 |
| MOL-E | −4.345 | −4.491 | −4.029 | −4.407 | −4.296 |
| MOL-F | −4.399 | −4.804 | −4.390 | −4.537 | −4.441 |
| MOL-G | −4.833 | −4.863 | −4.115 | −4.543 | −4.466 |
| MOL-H | −4.061 | −3.825 | −3.700 | −3.879 | −3.681 |
| MOL-I | −4.941 | −4.831 | −4.480 | −4.508 | −4.470 |
| MOL-J | −4.836 | −4.908 | −4.418 | −4.590 | −4.211 |
| NT | −5.622 | – | −5.545 | – | – |
| QT | −3.042 | −3.363 | – | – | – |

However, when QT is docked first in the protein and after NT is introduced into the Target-NT system, see column six, the LE is higher, representing a badly ligand-target interaction. As column five depicts, the same occurs when NT is docked first and after QT is introduced in the Target-QT system.

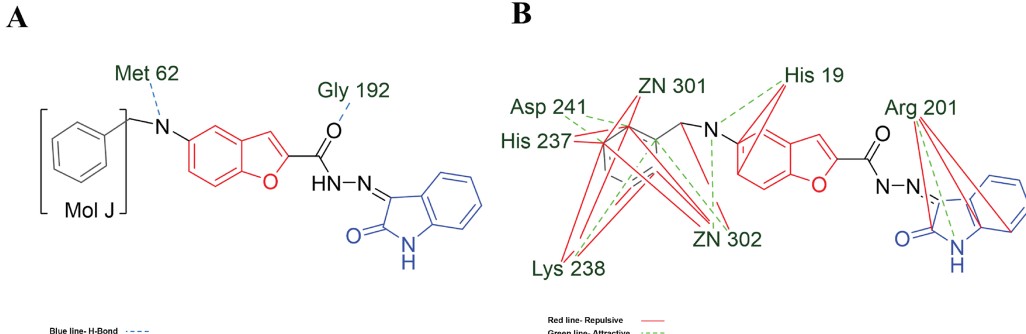

**Figure 7 MOL-J interacting with LpxC-NT (Target-NT) system.** (A) H-bonds, and (B) electrostatic interactions. Blue dotted lines depict the H-bond interactions, green dotted lines are the attractive, and red dotted lines are the repulsive electrostatic interactions. The Hex code for red color is #F50708 and for green colors are #50FF4F and #034F01.

**Table 5 Ligand efficiency (kcal/mol) for LpxC-NT system.**

| Name | VdW | Electro | H-bond* | LE |
|------|------|---------|---------|------|
| C-AZI | −0.424 | 12.283 | 0.000 | −2.487 |
| QT | −0.006 | −0.323 | 0.000 | −3.363 |
| MOL-H | −0.003 | −0.136 | 0.000 | −3.825 |
| C-CIP | −0.001 | −1.088 | 0.000 | −4.272 |
| MOL-E | 0.000 | 2.978 | 0.000 | −4.491 |
| MOL-A | −0.004 | 1.448 | 0.000 | −4.635 |
| IRB | 0.000 | 0.219 | 0.000 | −4.665 |
| JS_303 | −0.005 | 0.000 | 0.000 | −4.693 |
| MOL-F | −0.002 | 1.790 | 0.000 | −4.804 |
| MOL-B | −0.001 | 1.734 | 0.000 | −4.815 |
| MOL-D | −0.002 | 0.951 | 0.000 | −4.819 |
| BRB | 0.000 | 0.232 | 0.000 | −4.825 |
| MOL-I | −0.001 | 1.607 | 0.000 | −4.831 |
| MOL-G | −0.002 | 1.368 | 0.000 | −4.863 |
| IRA | 0.000 | −1.603 | 0.000 | −4.868 |
| MOL-C | −0.001 | 1.184 | 0.000 | −4.903 |
| MOL-J | −0.001 | 0.788 | 0.000 | −4.907 |
| BRA | −0.003 | −0.696 | 0.000 | −4.919 |

**Note:**
* H-bond means the hydrogen bond interactions, electro is the electrostatic interactions, VdW is the Van der Waals energy, and LE is the ligand efficiency (LE = E/No heavy atoms).

Table 3 shows the principal interaction energies for the target-NT assay. The cause that MOL E interacts better than the other designed molecules is not clearly explained. In this order, Fig. 5A shows the hydrogen bond interactions between MOL-E and Hfq-NT (target-NT), and Fig. 5B shows the electrostatic interactions.

In the case of hydrophobic surfaces, were analyzed the MOL-A in the Hfq, see Fig. 6A, and the MOL-E in the Hfq-NT system, see Fig. 6B.

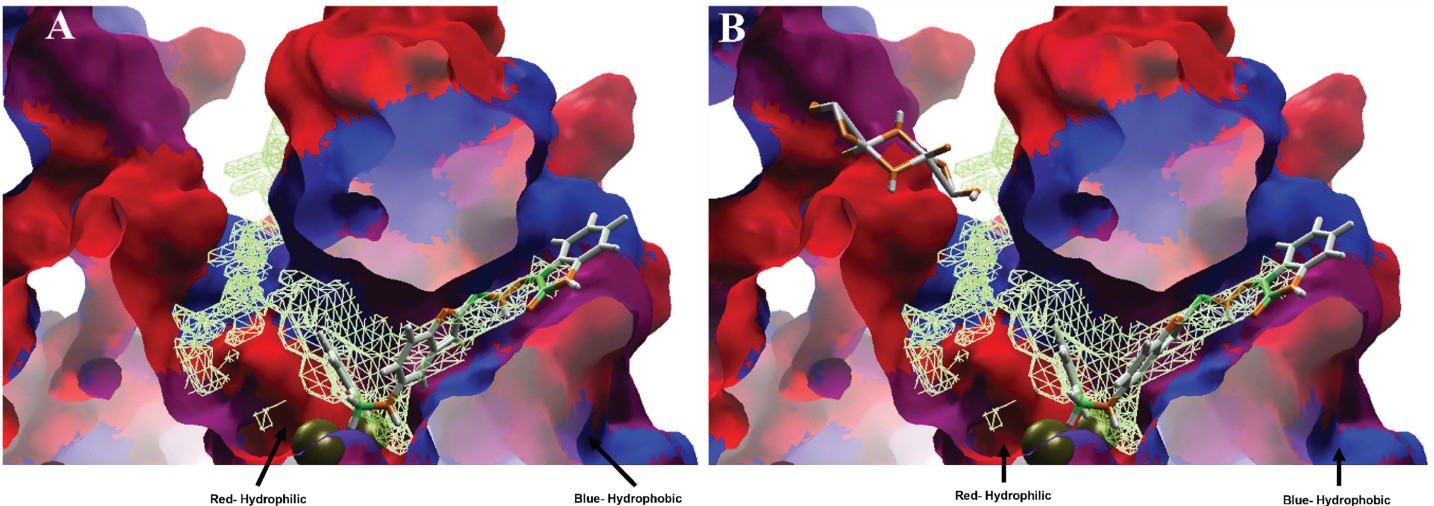

**Figure 8** **Hydrophobic interactions between protein LpxC with (A) MOL-I, and LpxC-NT with (B) MOL-J.** Blue surfaces represent hydrophobic sites, and red surfaces are hydrophilic zones. The Hex code for red color is #FE0021, and for green colors are #4BE34B and #A1D788.

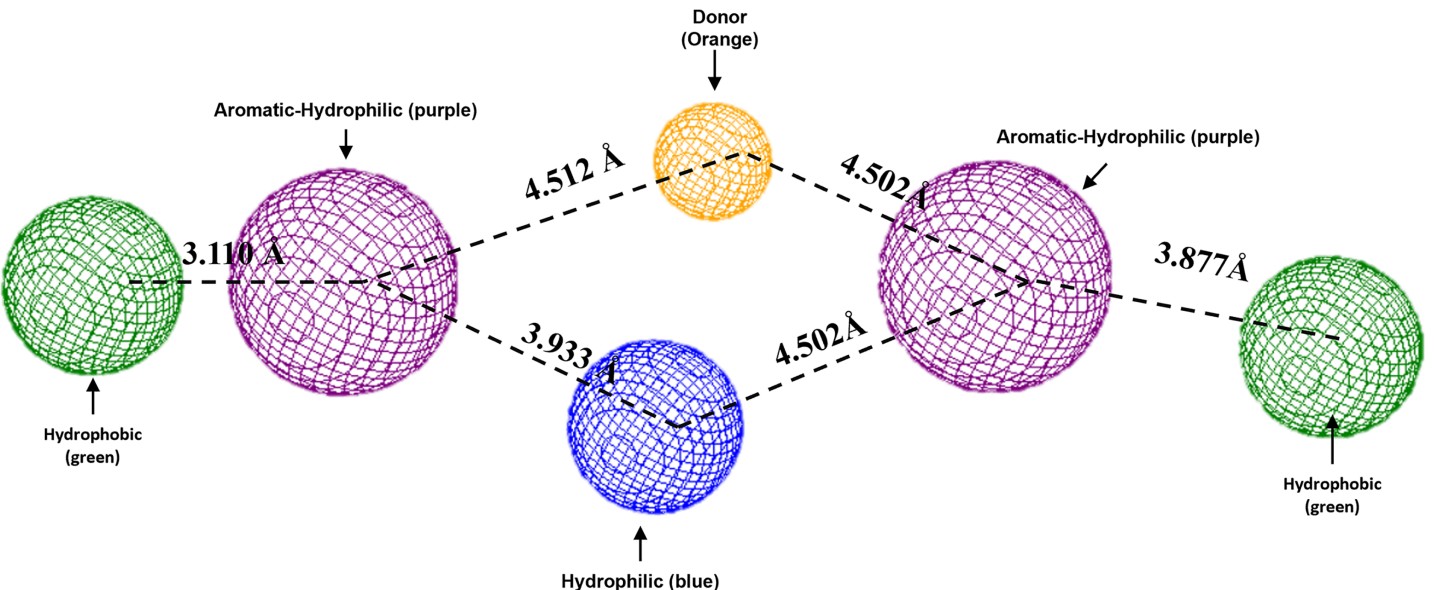

**Figure 9** **Pharmacophoric model for the Hfq protein of *P. aeruginosa*.** Green spheres represent the hydrophobic fragments, purple spheres represent the aromatic-hydrophilic fragments, blue spheres represent the hydrophilic fragments, and orange spheres represent the hydrogen donor fragments. The Hex code green color is #72B972.

Regarding the docking in LpxC, Table 4 shows that the influence of NT is similar to Hfq protein due to the presence of NT (coupling the designed molecules in the LpxC-NT). Above promotes a higher LE in molecules BRA, MOL-A, MOL-B, MOL-C, MOL-E, MOL-F, MOL-G, MOL-J (Figs. 7A and 7B), and QT than designed molecules coupled on free target (LpxC), see Table 4 column two and three.

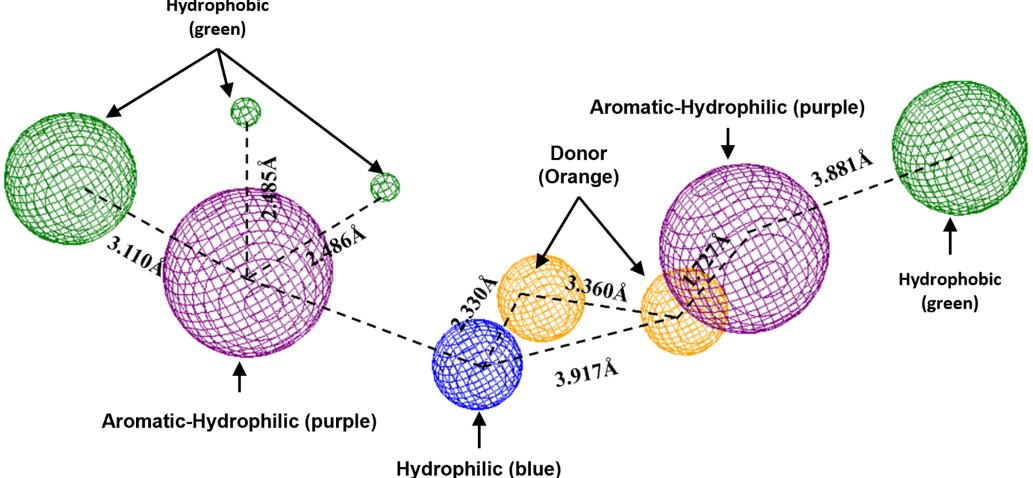

**Figure 10 Pharmacophoric model for LpxC protein.** Green spheres represent the hydrophobic fragments, purple spheres represent the aromatic-hydrophilic fragments, blue spheres represent the hydrophilic fragments, and orange spheres represent the hydrogen donor fragments. The Hex code green color is #72B972.

Analyzing the specific interactions of the studied molecules with the LpxC-NT (Table 5), the behavior is similar to Hfq protein, not being determinant Vdw, Electro of H-bond interactions to the ligand-target whole interaction.

Moreover, analyzing the hydrophobic surfaces of LpxC, Fig. 8A shows MOL-I docked into the free main cavity of LpxC, and Fig. 8B shows the interaction of MOL-J with the LpxC-NT system.

To conclude, the present study made a pharmacophoric model for each studied protein (Figs. 9 and 10). This study was based on the ligand interactions and proposed the main kind of fragments to try to inhibit this protein. Note that the green spheres depict the hydrophobic fragments, purple spheres are the aromatic-hydrophobic fragments, blue sphere is the hydrophilic fragments, and orange spheres are the hydrogen bond donor fragments.

# DISCUSSION

## Condensed Fukui function

In regard to condensed Fukui functions analysis, MOL A shows an electrophilic behavior with a value of 0.337 in the C29 and nucleophilic with 0.022 in C18. A radical-attack zone is less, with a value of 0.138 in C20. The reactive zones for (B–D) are in the C28, F-with around 0.337.

Molecule A tends to react with an electrophilic attack on the C29 atom (atom 28 in Molecule B–D). This zone corresponds to the structure of isatin in position 2 with a secondary carbonyl group. A possible radical attack occurs at the C20 position for this molecule, which is a secondary carbonyl linked to isatin and benzofuran. This behavior is similar in molecules (B–D).

## Molecular docking and pharmacophoric model

On the other hand, the influence of introducing the first NT in the target Hfq is benefic for the ligand-target interaction, such as with BRB, C-CIP, IRA, and MOL E, which the LE diminishes, which means a better ligand-target interaction. Despite MOL E not presenting better interaction than the referencing molecules (see Figs. 4 and 5, BRA, BRB, *Budhwani, Sharma & Kalyane, 2017*; IRA, and IRB, *Chemchem et al., 2020*), when it is docked with the presence of NT (Table 2, second column), their ligand efficiency is better. The ligand-target interactions beat the values for the commercial drugs (C-AZI and C-CIP), with a value of −4.321 kcal/mol. Note that there is a reported interaction between Hfq with $TiO_2$ NPs, but the importance of this as a nanocarrier is not highlighted (*Anupama et al., 2019*).

In regard to Table 2 results and analyzing the structures depicted in Fig. 1, Mol E presents two Flours substituting the benzene moiety, conferring this possible better interaction. MOL-E interacts specifically with Ser 6 and Gln 41 residues in terms of hydrogen bond interactions, see Fig. 5A. In the case of electrostatic interactions, Fig. 5B highlights that the NT confers higher attractive electrostatic interactions with MOL-E, also interacting more with the oxygens of the molecule.

Concerning hydrophobic interactions, the free-designed molecule (MOL-A) is introduced into the cavity, interacting more with the hydrophilic zone of MOL-A with the hydrophilic surfaces of Hfq (Fig. 6A). In the case of MOL-E (docked in the Hfq-NT system), the NT is introduced in the bottom of the cavity, interacting with a hydrophilic surface of Hfq. This first interaction promotes the designed molecule (MOL-E) docket out of the cavity, specifically on a hydrophilic surface (Fig. 6B).

On the other hand, in the case of the docking in LpxC, note that the ligand with the higher LE is the MOL-I (LE of −4.941 kcal/mol), and in the case of the LpxC-NT system, is MOL-I (LE of −4.908 kcal/mol). This means that the influence of NT in the target is favorable to the ligand-target interaction. Considering the structure of MOL-J, see Fig. 1, is the small of all the designed systems, which is a determinant of making a better LE. Despite the determinants of H-bond and Electro interactions, Fig. 7A shows that Met 62 and Gly 192 interact *via* H-bond with MOL-J, and the Zn co-factor is determined in the electrostatic interactions of MOL-J, see Fig. 7B.

Moreover, analyzing the hydrophobic surfaces of LpxC, Fig. 8A shows MOL-I docked into the free main cavity of LpxC, which interacts mainly with hydrophobic surfaces of the protein and covers the surface of the studied cavity. Figure 8B shows the interaction of MOL-J with the LpxC-NT system, which exposes the influence of NT on the system. This figure clearly shows that NT is introduced in the deep zone of the cavity, promoting a similar interaction but with higher LE.

To conclude the present study, the pharmacophoric model was formulated. Figure 9 shows the pharmacophoric model for the Hfq protein of *P. aeruginosa*, demonstrating that six fragments are necessary to dock some molecules better to this protein. Specifically, it needs two hydrophobic fragments at both sites of the designed molecule (green spheres), followed by two aromatic-hydrophobic fragments (purple spheres). Describing the

model's center is necessary for one hydrophilic (blue spheres) and other hydrogen bond donor fragment (orange sphere).

In the case of the pharmacophoric model for LpxC protein, it needs nine fragments to be docked with some designed molecules, see Fig. 10. The first two are similar to the Hfq model, being two hydrophobic (green spheres) sites at the sites of the molecule. The following fragments are also two aromatic-hydrophobic sites (green spheres). The left side of the molecule needs two small hydrophilic fragments, and the molecule's center should need two hydrogen bond donor fragments (orange spheres) and one hydrophilic site (blue sphere).

## CONCLUSIONS

Ten modified benzofuran-isatin molecules were studied through computational simulation, using density functional theory (DFT) and molecular docking assays against Hfq and LpxC (proteins of *P. aeruginosa*).

The control molecules BRA, BRB, IRA, and IRB generally have a better ligand efficiency (most negative than −4.000 kcal/mol). This is due to the ligand efficiency, energy was considered as the number of atoms per molecule when determining the binding strength between the ligand and the target. The results show that commercial drugs C-CP and C-AZI against Hfq present −2.126 and 3.744 kcal/mol, respectively, compared with the best-designed molecule with −4.265 kcal/mol (MOL-A).

However, we highlight that the influence of NT promotes a better interaction of some molecules; for example, MOL-E generates a better interaction by 0.219 kcal/mol (from −4.102 to −4.321 kcal/mol) when NT is introduced in Hfq, forming the system Hfq-NT (Target-NT). Similar behavior is observed in the LpxC target, in which MOL-J is better at 0.072 kcal/mol (from −4.836 to −4.908 kcal/mol) in the presence of NT in the target.

The molecules (A, E, I, and J) show better activity against LpxC and Hfq protein. The D and G molecules are favored with the introduction of NT and QT, precisely when NT enters first, obtaining values of −4.028 and −4.314 kcal/mol, respectively.

Finally, two pharmacophoric models for Hfq and LpxC implicate hydrophobic and aromatic-hydrophobic fragments. All proposed molecules (A–J) have more significant inhibition benefits than commercial drugs C-CIP and C-AZI. In other words, they have greater bacterial inactivation capacity.

### Funding

Verónica-Castro-Velazquez received a scholarship from CONACYT (No. 951424). This work was supported by the CNS-IPICyT (Centro Nacional de Supercómputo-Instituto Potosino de Investigación Científica y Tecnológica A.C.), by the Laboratorio Nacional de Caracterizacion de Propiedades Fisicoquimicas y Estructura Molecular (UG-UAA-CONACYT, Project: 123732), and by the Laboratorio de Caracterización Molecular de

Biosistemas y Nanocompuestos (LACAMBION) by the computer time provided at "La Biznaga" HPC. Also, CONACYT provided financial support granted to the project "Materiales activos para agricultura sustentable, 101703—Ciencia de Frontera, 2019." The funders had no role in study design, data collection and analysis, decision to publish, or preparation of the manuscript.

## Grant Disclosures

The following grant information was disclosed by the authors:
CONACYT: 951424.
CNS-IPICyT (Centro Nacional de Supercómputo-Instituto Potosino de Investigación Científica y Tecnológica A.C.).
Laboratorio Nacional de Caracterizacion de Propiedades Fisicoquimicas y Estructura Molecular: 123732.
Laboratorio de Caracterización Molecular de Biosistemas y Nanocompuestos (LACAMBION).
Materiales Activos Para Agricultura Sustentable: 101703—Ciencia de Frontera: 2019.

## Competing Interests

The authors declare that they have no competing interests.

## Author Contributions

- Verónica Castro-Velázquez performed the experiments, performed the computation work, prepared figures and/or tables, and approved the final draft.
- Erik Díaz-Cervantes conceived and designed the experiments, performed the experiments, analyzed the data, authored or reviewed drafts of the article, and approved the final draft.
- Vicente Rodríguez-González analyzed the data, prepared figures and/or tables, authored or reviewed drafts of the article, and approved the final draft.
- Carlos J. Cortés-García analyzed the data, prepared figures and/or tables, and approved the final draft.

## Data Availability

The raw data is available in the Supplemental File.

## Supplemental Information

Supplemental information for this article can be found online at http://dx.doi.org/10.7717/peerj-pchem.27#supplemental-information.

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
