# Peer review of "In-silico* assay of a dosing vehicle based on chitosan-TiO₂ and modified benzofuran-isatin molecules against *Pseudomonas aeruginosa"

_PeerJ Physical Chemistry, doi:10.7717/peerj-pchem.27_

## Round 0.1 · original submission · Minor Revisions

Please address the comments raised by the reviewers, in particular with respect to the experimental design. The suggested citations may be included if you deem them relevant.

Reviewer 1 ·

Basic reporting

Overall the manuscript is very well structured.

Experimental design

- In the Molecular Docking and pharmacophoric model, some residues are missing in crystallographic structure with PDB coded 5I21and 4J3D. Did the authors add it? In that case, please clarify how they did it.
- What method was used to perform the molecular docking and validate the docking process?
- What control compounds are used? I could not locate one.

Validity of the findings

'no comment'

Additional comments

'no comment'

Annotated reviews are not available for download in order to protect the identity of reviewers who chose to remain anonymous.

·

Basic reporting

no

Experimental design

The research question and the aim are not clearly defined

Validity of the findings

The novelty of this study is not fully addressed

·

Basic reporting

The article entitled “In-silico assay of a dosing vehicle based on chitosan-TiO2 and modified benzofuran-isatin molecules against Pseudomonas aeruginosa” This paper looks scientifically sound, and interesting to the readers of Peer J Physical Chemistry. After addressing the few recommendations made in this article. The manuscript may be accepted for publication in this journal. A few minor corrections are suggested:

Experimental design

NA

Validity of the findings

The article entitled “In-silico assay of a dosing vehicle based on chitosan-TiO2 and modified benzofuran-isatin molecules against Pseudomonas aeruginosa” This paper looks scientifically sound, and interesting to the readers of Peer J Physical Chemistry. After addressing the few recommendations made in this article. The manuscript may be accepted for publication in this journal. A few minor corrections are suggested:

Additional comments

Manuscript No.: Peer J-2022.
Recommendation: Minor revision.
Comments:
The article entitled “In-silico assay of a dosing vehicle based on chitosan-TiO2 and modified benzofuran-isatin molecules against Pseudomonas aeruginosa” This paper looks scientifically sound, and interesting to the readers of Peer J Physical Chemistry. After addressing the few recommendations made in this article. The manuscript may be accepted for publication in this journal. A few minor corrections are suggested:
1. The abstract is too long; I ask the authors to shorten it and leave only the most important information.
2. Give the proper reason behind the utility of B3LYP/6-31+G(d,p) and M06-2x. A justification must be provided.
3. To put the study in its state of the art, the authors must update the Introduction section, by recent works found in the literature. Introduction section should be revised. Some references which should be added: https://doi.org/10.1016/j.compbiolchem.2020.107311, DOI:10.1016/j.saa.2013.11.001, https://doi.org/10.1016/j.molstruc.2021.131083, https://doi.org/10.1016/j.jmrt.2019.11.045.
4. Why is the choice of the method employed in this study? Proper citation for the used basis set should be introduce.
5. The abbreviations in the manuscript should be provided with the full name before use, e.g "NBO”, "DFT", etc.
6. How do the author chose different proteins in molecular docking? Explain more and compare the results found with these found in the literature.
7. The manuscript should be thoroughly scrutinized for punctuations/grammatical errors.
Remark: Accept with minor revision

---

## Round 0.2 · accepted · Accept

Thank you for addressing the points raised by the reviewers. After assessing the revised manuscript, I am happy to accept the paper for publication.